# Analysis of the Impact of Communication Campaigns under the Project “Syphilis No”: A National Tool for Inducing and Promoting Health

**DOI:** 10.3390/ijerph192315884

**Published:** 2022-11-29

**Authors:** Jordana Crislayne de Lima Paiva, Sara Dias-Trindade, Mario Orestes Aguirre Gonzalez, Daniele Montenegro da Silva Barros, Pablo Holanda Cardoso, Pedro Henrique Castro Bezerra, Thaisa Gois Farias de Moura Santos Lima, Juciano de Sousa Lacerda, Lilian Carla Muneiro, Aliete Cunha-Oliveira, Ana Paula Camarneiro, Ingridy Marina Pierre Barbalho, Ricardo Alexsandro de Medeiros Valentim

**Affiliations:** 1Laboratory of Technological Innovation in Health, Federal University of Rio Grande do Norte, Natal 59010-090, Brazil; 2Centre for Interdisciplinary Studies (CEIS20), University of Porto, 4099-002 Porto, Portugal; 3Postgraduate Program in Production Engineering, Federal University of Rio Grande do Norte, Natal 59078-970, Brazil; 4Federal Senate of Brazil, Brasilia 70160-900, Brazil; 5Postgraduate Program in Media Studies, Federal University of Rio Grande do Norte, Natal 59078-970, Brazil; 6Health Sciences Research Unit: Nursing (UICISA:E), Nursing School of Coimbra (ESEnfC), CEIS20-UC, 3000-457 Coimbra, Portugal; 7Health Sciences Research Unit: Nursing (UICISA:E), Nursing School of Coimbra (ESEnfC), 3000-076 Coimbra, Portugal

**Keywords:** public health promotion, health communication campaigns, syphilis, performance indicators

## Abstract

Syphilis is increasingly prevalent around the world as a result of complex factors. In Brazil, the government declared a syphilis epidemic in 2016 and then set a strategic agenda to respond to this serious public health problem. In a joint effort, Brazil’s Federal Court of Accounts (TCU) recommended that novel and diversified health communication strategies should be developed, which the “Syphilis No” project (SNP) later conducted through nationwide mass communication campaigns. We performed exploratory data analysis to identify and understand the results of three health communication campaigns by considering syphilis data trends in Brazil. The SNP, by using traditional and innovative means of communication, focused on multiple target audiences to encourage behavior changes through awareness and syphilis knowledge acquisition via the internet. In addition, the SNP disseminated information on syphilis testing, prevention, and treatment through social media and multiple media outlets. We observed that the period of the health campaigns corresponded to the period when the syphilis testing uptake increased and the number of reported cases dropped. Thus, our findings indicate that public health responses could substantially benefit from the use of health communication campaigns as a tool for health promotion, education, and transformation.

## 1. Introduction

Syphilis is a serious public health issue. It is a sexually transmitted infection (STI) caused by the bacterium Treponema pallidum and is curable, systemic, and highly contagious. Untreated syphilis can progress to more severe levels, especially compromising the nervous and cardiovascular systems [1,2].

Despite the availability of cheap and effective antibiotic therapy, syphilis remains a prevalent disease in developing countries and has re-emerged as a public health threat in developed countries [3,4,5]. Syphilis has an estimated global prevalence of 36 million cases and an incidence of over 11 million cases annually [6].

In Brazil, according to official data from the Epidemiological Bulletin of the Ministry of Health of Brazil (MH), in 2020, the incidence rate (per 100,000 inhabitants) of acquired syphilis was 54.5; this rate was 21.6 for syphilis in pregnant women (per 1000 live births); and 7.7 for congenital syphilis in children under one (per 1000 live births) [7]. In Brazil, congenital syphilis became a disease of compulsory notification in 2005, and acquired syphilis reached the same level in 2010 [4].

In 2016, the World Health Organization (WHO) launched the Global Health Sector Strategy on Sexually Transmitted Infections. They set an impact target to reduce syphilis infections by 90% globally between 2018 and 2030 [8]. In the same year, the Brazilian government declared an epidemic of syphilis in the country and drew up an agenda of strategic action to combat the infection. The following year (2017), an extensive audit carried out by the Brazilian Court of Auditors (TCU) found that the syphilis control policy was deficient across the Brazilian territory. This audit resulted in Judgment No. 2019/2017 [9], which recommended new and diversified health communication strategies, targeted to specific audiences, for example, teenagers, sex workers, and populations without internet access. The other TCU recommendations for the effective communication and control of syphilis in Brazil included other audiences such as health professionals, managers, pregnant women, sexual partners, etc. The main goal was to enhance the dissemination of information about the problem and reinforce the impact of prevention measures among the population, particularly key populations.

The “Agenda of Strategic Actions for the Reduction of Syphilis in Brazil” was developed by the Brazilian Ministry of Health. This resulted in the implementation of a research and intervention project with innovative strategies applied for the response to the syphilis epidemic in Brazil [10]. For the project’s management, the MH signed a technical-scientific partnership with the Federal University of Rio Grande do Norte (UFRN). Through this partnership, the “Syphilis No” Project was prepared and developed [1,2,4,11,12].

Among the objectives of the “Syphilis No” project, especially in the area of communication, are: (1) operationalizing a communicative ecosystem with broad and solid relationships between the areas of communication and education to address the issue of syphilis; and (2) using technological mediation with a focus on education and communication to disseminate knowledge aimed at professionals, managers, users, key populations, and the general population [13].

To achieve these goals, the project developed activities in the public health communication area through mass communication campaigns [2]. The campaigns were built and widely used for the dissemination of awareness messages, a factor that contributed to expanding information about social agendas, and also for persuasion and encouragement in the adoption of attitudes and behaviors related to health by society [11,14].

In the field of public health communication, especially for syphilis awareness, social campaigns are acknowledged as a promising strategy that can lead to improved health by changing the behavior of individuals [15,16]. In addition, communication can be applied as an intrinsic strategy for health promotion. It is considered an important public health intervention tool capable of inducing the prevention and control of infectious diseases [17]. Thus, this study starts with the hypothesis that the three communication campaigns developed by the “Syphilis No” project had a positive and considerable impact on the syphilis scenario in the national territory in the period from 2018 to 2020. In order to guide the research, the following questions were defined: (i) Did the communication campaigns of the “Syphilis No” project stimulate the interest of the general population on the subject? and (ii) Did the campaigns induce an increase in testing?

Given the contexts addressed, this study aims to describe the impact of the public health communication campaigns of the “Syphilis No” project on the approach to syphilis in Brazil and analyze how these campaigns contributed to inducing behavioral changes that resulted in the improvement of syphilis indicators in Brazil, considering the data from 2015 to 2020.

## 2. Materials and Methods

The exploratory data analysis method was chosen with the goal of identifying, knowing, and understanding the trends presented in the data according to the steps suggested by Cox [18]: (i) Data are collected; (ii) What data types are there? (iii) What do the data look like? (iv) Is there anything odd about the data? (v) Which families do the data come from? and (vi) Which tests can be performed?

Figure 1 presents the exploratory analysis process that consists of classifying the diversity of data present in the sample universe, understanding them, and deciding what information is relevant and how it can be presented. In addition, behavioral familiarities are identified and the analysis process is replicated and tested in a variety of scenarios.

When applying the process suggested by Cox [18] in this research, the following steps were followed: (i) collection of data and epidemiological information on syphilis in Brazil; (ii) identification of official data from government agencies on syphilis in Brazil (Brazilian Ministry of Health and Brazilian Federal Court of Auditors); (iii) analysis of the data and information collected; (iv) identification of the problem, considering that the government carried out communication tasks with the objective of combating syphilis but the number of cases had grown; (v) mapping of different family/data sources on the research topic; and (vi) discussion about the research and its guiding questions.

The “Syphilis No” project is classified as the largest research project for inducing public policies to fight syphilis in Brazil. The project is composed of several thematic axes in relation to this health issue. We decided to study the communication axis. Among several strategies, we decided to analyze the public communication campaigns that were carried out in the project due to their behavioral familiarity and important level of detail. Public data on campaign formats make it possible to replicate them for distinct locations or diseases.

### 2.1. Data Identification, Extraction, and Analysis

The data were collected from 2015 to 2020. The objective was to analyze the number of diseases, the number of diagnostic tests performed, and the number of searches on ‘syphilis’ in the Google search engine. Based on this period, data were extracted from (i) the 2021 Syphilis Epidemiological Bulletin published by the Brazilian Ministry of Health [7]. The number of cases of acquired syphilis, syphilis in pregnant women, and congenital syphilis were considered; (ii) research on the scope of public health campaigns based on multidimensional aspects [11] from 2015 to 2019 and the SUS Outpatient Information System (SIA/SUS) for 2020; and (iii) Google Trends to gauge the population’s interest in ‘syphilis’, that is, in the search for more knowledge and information on the theme.

For the analysis, data from the campaigns were also used, which were extracted from the “Syphilis No” project, aimed at understanding the main methods, types of communication vehicles used, and territories reached by the three campaigns analyzed by the present work. Finally, an individual analysis of the data obtained was carried out, and then, a descriptive analysis of the information in terms of the methodology used was carried out. Thus, the final study resulted in a solid analysis of the subject.

### 2.2. Campaign Design

The advertising campaigns analyzed by the present study were carried out from 2018 to 2021 in the context of the “Syphilis No” project. Table 1 shows the year, theme, focus, and target audience of these campaigns.

The first campaign took place in 2018 and had a conceptual characteristic, namely it allowed for the exploration of the synthesis brand developed for the project based on its extensive official name. There was a discussion and a study to shorten the official name of the project “Applied Research for Intelligent Integration Oriented to Strengthening Care Networks for Rapid Response to Syphilis” to “Syphilis No” (Portuguese term “Sífilis Não”) for the creation of the brand, as represented in Figure 2 and Figure 3.

The definition of the campaign theme was “Syphilis No! #TestTreatCure” (term in Portuguese #TesteTreateCure), which represents the interruption of the disease transmission chain. The project’s brand was created (Figure 3), with an emphasis on the following characteristics: (1) the colors represent the different groups to be reached by the dissemination; (2) Brazil’s map, alluding to the national scope of the project; (3) the use of the hashtag “#TestTreatCure”, representing the entire line of care needed; and (4) the illustration of a hand as a stop or interruption symbol.

The second campaign was carried out through an advertising agency hired through a public selection process that ran between 2018 and 2019. The campaign addressed a variety of topics and formats on the prevention, testing, and treatment of syphilis. The target audience was expanded to include women, pregnant women, men, managers, and health professionals in priority populations and key populations (as shown in Table 2). The messages were built using non-imposing language, focusing on affection and the dimension of self-care and care for others. The slogan was “Remember to take care of yourself: test, treat and cure”.

The second campaign was characterized by a combination of production, circulation, and consumption strategies in cross-media [21,22] and trans-media formats [23,24,25]. The main aspect of this campaign was the greater diversity of the means and vehicles of communication, as illustrated by the word cloud in Figure 4, and it was distributed throughout the national territory. The second campaign was a massive undertaking as it was developed for the Brazilian national media and soon had universal coverage on the theme of syphilis. Thus, it is certainly the largest public health communication campaign dedicated exclusively to syphilis in the last 20 years in Brazil.

In the third campaign, which took place between 2020 and 2021, the starting point was an individual statement “I know” (Portuguese term “Eu sei”) for the question “Do you know?” (Portuguese term “Você sabe?”). The idea aimed to generate curiosity and inferences regarding care and stimulate dialogue on the theme. Figure 5 represents the visual identity developed for the campaign.

The main focus of this campaign was to strengthen the discourse on syphilis prevention by encouraging diagnostic testing. Organic communication activities were carried out [26] with strategies aimed at digital communication platforms. They were aimed at the general population, young people, sexual partners, drug and alcohol users, the elderly, health professionals, pregnant women, Indigenous people, homeless people, and managers.

## 3. Results

### 3.1. Campaign 1 (2018)

The communication campaigns of the “Syphilis No” project had different characteristics. Campaign 1 was fundamental to establishing the identity of syphilis in Brazil. The brand created by the researchers of the “Syphilis No” project was used as an icon in communication strategies to combat the disease in several regions of the country and became a symbol of combating syphilis in Brazil. For example, the red ribbon symbol was used to talk about HIV/AIDS.

Entities with no direct link to the project used the brand in Figure 2 in their own strategies. Table 3 describes examples of the project’s visual identity not being fully applied or adapted in the states of São Paulo, Tocantins, Minas Gerais, and Rio Grande do Norte in state strategies for communication about syphilis.

Another significant result of the campaign was the creation of project profiles on the main social networks (Facebook, Instagram, Twitter, and YouTube) and the creation of themed pieces for the Carnival period (Figure 6). Carnival marches (popular songs), historically known in discursive operations of deconstruction and reconstruction, were amended with new verses, whose composition provoked the attention of the target audience to the problem of syphilis. To improve the understanding of the elements in Figure 6, the English translations are as follows:Figure 6A: Oh, clear the way because I want to get tested;Figure 6B: Doctor, I’m not wrong, I’m going to take the test because I love myself;Figure 6C: It’s those who use condoms that we like the most;Figure 6D: The waters will roll and from syphilis you can be cured;Figure 6E: Pierrot in love used a condom and doesn’t cry for Columbine;Figure 6F: Are you going to cross the Sahara Desert? Always take a condom with you.

It is important to highlight that Carnival is the biggest popular festival in Brazilian culture, is internationally known, and is also a national holiday. In this context, six pieces were produced and disseminated through the project’s social networks. As a result, at the end of the campaign, 144 (one hundred and forty-four) release strategies were recorded.

### 3.2. Campaign 2 (2018–2019)

Campaign 2 was developed through an advertising agency and broadcast nationwide. The agency used various communicative strategies disseminated through different media and communication vehicles [27]. Among the main types of media, the following stand out: digital media, social networks, influencers, external media, print, radio, streaming platforms, television stations (TV), and awareness blitzes.

The advertising messages were created aimed at disseminating information about syphilis, encouraging diagnostic testing, informing about the existence and relevance of undergoing treatment to cure the infection, and changing the healthcare landscape.

On TV, the videos “Teste, Trate e Cure” (https://www.youtube.com/watch?v=CtjSTx-VX4g), “Lembrete aos Jovens” (https://www.youtube.com/watch?v=yoY1V5um97w), “Lembrete às Gestantes e Parcerias Sexuais” (https://www.youtube.com/watch?v=0dcDAVQEBmM), and “Lembrete para Festas” (https://www.youtube.com/watch?v=ZcGuGYTjIMw) (in 30” format) were linked to the four main broadcasters in the country, and on closed TV on AMC Brasil and HBO, videos on congenital syphilis and acquired syphilis were broadcast, aimed at young audiences aged 15 to 24 years old, gay people, men that have sex with men (MSM), and men aged 20 to 35 years old, as shown in Table 4.

Still in the TV context, merchandising and statement strategies were used. For this purpose, 44 appeals using 60” videos were used, along with 5 newsletters and 170 insertions into 30 LGBT commercial videos. Another important strategy was on TV Rede Meio Norte, one of the largest communication groups in Piauí, where 84 merchants and 260 commercials were broadcast.

As shown in Table 5, Campaign 2 reached an average of 16,374,777 TV viewers, with a total radio audience impact of 4,141,527 times. In print media, 10,800,000 products were produced including posters (193,500), stickers (770,000), and folders (770,000) for health professionals and the general public. A total of 38,000 informative booklets were also produced and 8000 condoms were distributed in awareness-raising activities. Figure 6 shows a summary of all strategies carried out during Campaign 2.

The campaign also used educational-communicative strategies focused on education and communication on the subject. Moreover, it developed specific products for each population, segmented message, and media vehicle. The objective was to increase adherence to the disseminated content through positive, affirmative, and engaging communication, which highlighted the responsibility and role of each citizen in relation to the care and action that must be taken to reduce syphilis cases in Brazil.

The quality of the material produced and its innovations in terms of public health communication stand out. Three videos were created (“Without wavering ” (https://www.youtube.com/watch?v=lmTzOrpAc70), “The Matter That Saves Lives” (https://www.youtube.com/watch?v=kr12Uq6ce9w), and “Syphilis Game Show” (https://www.youtube.com/watch?v=0TTr4JJeQlM), which are all available on the project’s YouTube channel) and received eight medals in the Columnists Award. The Columnists Award has been taking place since 1967 and is the oldest and most traditional communication and marketing award in Brazil. In addition, it is considered the highest award in Brazilian advertising. The objective is to highlight the most notable marketing communication strategies carried out by companies and professionals.

### 3.3. Campaign 3 (2020–2021)

Campaign 3 maintained the communicative characteristics of the previous campaigns. There was a variety of target audiences with the use of positive and targeted messages. This campaign stands out due to the fact that the personas (fictional characters representing the target audience) were not actors but real-life individuals. For example, Figure 7C, which was aimed at communicating the message to young couples, shows a real-life couple; Figure 7D, which was aimed at the gay community, shows a gay man; and Figure 7F, which was aimed at managers in the healthcare domain, shows an image of a real-life healthcare manager. This format allows the greater identification of the spectators with their peers. Figure 7 represents some of the pieces produced for Campaign 3. When analyzing them, one can see the diversity of the public intended to be reached: health professionals, pregnant women, Black women, men, families, young people, managers, the Black population, populations in situations of social vulnerability, the homeless population, Indigenous people, drug and alcohol users, and sex workers, among others. To improve the understanding of the elements in Figure 7, the English translations are as follows:Figure 7A: Syphilis. I know. Do you know? Syphilis doesn’t have to be taboo. Man, your health is important. Take the quick test today. Learn more at sifilisnao.com.br;Figure 7B: Syphilis. I know. Do you know? Mothers Day. The greatest gift for a mother is to have her child healthy. Stay tuned, moms: during pregnancy, don’t forget to perform prenatal care and take your quick syphilis test. Learn more at sifilisnao.com.br;Figure 7C: Syphilis. I know. Do you know? Drag to the side and know every symptom of syphilis!;Figure 7D: Syphilis. I know. Do you know? This carnival redouble care. Avoid agglomerations and have sex with the use of condoms. Syphilis isn’t over either. Learn more at sifilisnao.com.br;Figure 7E: Syphilis. I know. Do you know? Click the button and send this post to someone you love. Taking care of health is also a demonstration of love!;Figure 7F: Syphilis. I know. Do you know? Manager, I have a message for you! Take care of the health of the population. Offer the test, treat and cure! Learn more at sifilisnao.com.br.

The content was distributed organically through the official social networks of the project “Syphilis No” (*@sifilisnao* on Instagram and “Sífilis Não” on *Facebook* and *Twitter*). Through these channels, content produced on prevention measures, rapid testing, guidelines for sexual partnerships, and symptoms of syphilis, among other topics, were made available.

This campaign was able to reach 41,644 accounts and made 71,592 impressions. The “reach” means the number of unique accounts that have seen a post at least once. On the other hand, “impressions” is the number of views, which can include multiple views by the same user.

## 4. Discussion

Health communication carried out by public authorities is a fundamental element in the response to any event that may threaten human life [28]. The messages must ensure that each individual (i) understands the information; (ii) recognizes the information and its applications; (iii) assumes that they are exposed to greater risk if they do not adopt protective measures; (iv) decides the need to act based on the information received; (v) understands what action must be taken; and (vi) is able to act [16,29].

Investment in public health campaigns is one of the most effective methods for giving an issue relevance in the media agenda, public agenda, and public health agenda, and can cause a sense of uncertainty and a need for the individual subject to act [30]. In this context, one of the objectives of the campaigns carried out for the “Syphilis No” project was to increase national visibility in the media to promote the public health agenda and the transformation of the syphilis problem in the country. This transformation took place via stimulation of behavioral changes through awareness and subsequent searches for information on the internet by the general population and key and priority populations on topics such as the testing, prevention, and treatment of syphilis. The different types of media products in the campaigns associated with events and the distribution of materials associated with the theme are recognized strategies in terms of efficiency [31].

As a complement to the analysis of the impact of the campaigns carried out, the epidemiological indicators of acquired syphilis in pregnant women and congenital syphilis in Brazil showed an increasing number of cases since 2011, a trend that began to change from 2018 onward [32]. The number of cases of acquired syphilis in 2018 was 159,237. In 2020, that number dropped to 115,371, representing a decrease of 27.55%. Regarding congenital syphilis, the number of cases in 2018 was 26,464. In 2020, this figure decreased to 22,065, representing a decrease of 16.62%. The same pattern occurred for the number of syphilis cases in pregnant women, which in 2018 was 63,250 and in 2020 was 61,441, representing a decrease of 2.86%. Figure 8 shows the number of cases of acquired syphilis, congenital syphilis, and syphilis in pregnant women from 2010 to 2020. In this context, the content of the campaigns carried out during this period may have reached a large part of the target population and, consequently, influenced the reduction of cases of syphilis in Brazil.

Another relevant aspect for analyzing the impact of the campaigns carried out is the investigation of the types of web searches conducted. Between 2015 and 2020, Google Trends data on searches for the term “syphilis” in Brazil show a growing trend, which illustrates an increase in the number of Google searches over time.

In Figure 9A, the blue line represents searches performed on Google and the dotted line represents the trend in searches performed on Google by the general population. The values shown in Figure 9A are not absolute numbers of searches. Google assigns a value of 100 to the highest number of searches for the month and a proportional value to others. It is possible to see in the graph that there was a large volume of research in the year in which the epidemic was declared in 2016, and the highest volume of searches was in the year in which the communication campaigns developed within the scope of the SNP began, that is, in 2018, 2019, and 2020. The behavior of carrying out internet searches on the topic of syphilis may reflect the feeling of vulnerability [33] that the subject feels when faced with news regarding the need to do something to reduce their degree of uncertainty [30].

Another important point is the increase in the number of diagnostic tests. A study on the reach of public health campaigns based on multidimensional aspects by de Morais Pinto et al. [11] considered these indicators of performance: serological tests for syphilis, drug distribution (Penicillin G benzathine), and case notification rates in primary healthcare (PHC). All these data were from 2015 to 2019 and were collected from the SIA/SUS database. In addition, the data for 2020, the year of the last communication campaign analyzed, were also taken from SIA/SUS database. The data collected were related to population data, with the rate calculated per 100,000 inhabitants, and are plotted in Figure 9B.

According to the data, there has been an increase in diagnostic testing since 2015, with a significant increase occurring in 2018, even when accounting for the country’s population growth. When compared to 2015, this shows that 3.39 times more individuals were tested in 2018, 4.75 times more in 2019, and 22.96 times more in 2020 across the country based on public data released by the Ministry of Health. This aspect demonstrates an important relationship between the public health communication interventions carried out for the “Syphilis No” project and the increase in testing throughout the country. It is noteworthy that testing is one of the premises for the diagnosis, treatment, and cure of syphilis, and all these topics were addressed in various ways in the campaigns developed for the “Syphilis No” project. Therefore, communication was an important intervention strategy in this project, as it disseminated information on the theme of syphilis as part of the public health agenda to more than five thousand municipalities in Brazil. In addition, it reached the key populations recommended by the WHO.

The period in which the campaigns were carried out and broadcast corresponded to the period in which there was a positive change in people’s attitudes toward syphilis in Brazil. The data prove the increase in interest in the topic “syphilis” by the population, the volume of tests carried out, and the reduction in the number of cases reported across the country. This suggests that the campaigns constituted public health interventions necessary for changes to occur, as they ensured that the syphilis topic remained on the public agenda between 2018 and 2020. This aspect caused the population to become more interested in information about the disease and its consequences and corroborates the important changes in the trend in syphilis cases in the country, which had increased continuously for more than a decade, but which began to change after the beginning of the “Syphilis No” project.

## 5. Conclusions

It is important to highlight that the pandemic may have affected the control of syphilis notifications. However, the significant increase in testing, particularly in 2020 (more than 22 times compared to 2015) shows that there was effectiveness in patient care. Thus, syphilis was not neglected in this period in Brazil. Further studies on this topic will be necessary to better understand the effects of the pandemic and its relationship with sexually transmitted infections. It is noteworthy that during the pandemic, there was a renegotiation of the strategic agenda to combat syphilis in Brazil and other training strategies for health professionals and supporters in the territory, promoted by the Brazilian Ministry of Health through technical cooperation with the “Syphilis No” project. This point demonstrates that even during the pandemic, Brazil acted to maintain the public health agenda’s approach to syphilis. This also helps to explain the level of resilience of the health system, which may have also contributed to the reduction in cases.

As a result of innovative, transdisciplinary, intersectoral, and interfederative planning, it is possible to conclude that communication campaigns are effective health induction and promotion tools in the context of public health. However, campaigns cannot be stated as being the only reason for the changes in trends in syphilis cases in Brazil. It is recommended to continue research in this area of study, expanding it to other regions or diseases to investigate whether the methodology used to disseminate information for the “Syphilis No” project increased the interest of the general population and healthcare professionals on the subject and improved the relevant health issue.

## Figures and Tables

**Figure 1 ijerph-19-15884-f001:**
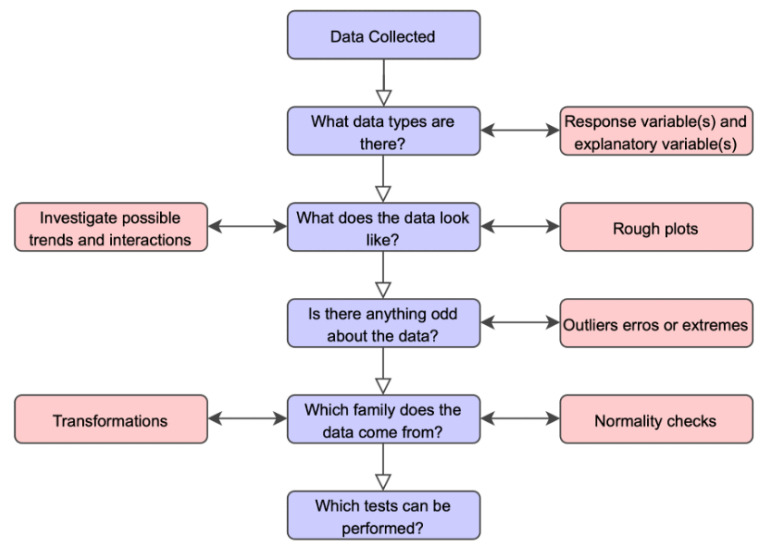
Exploratory Data Analysis (EDA) process. **Source**: Cox [18].

**Figure 2 ijerph-19-15884-f002:**
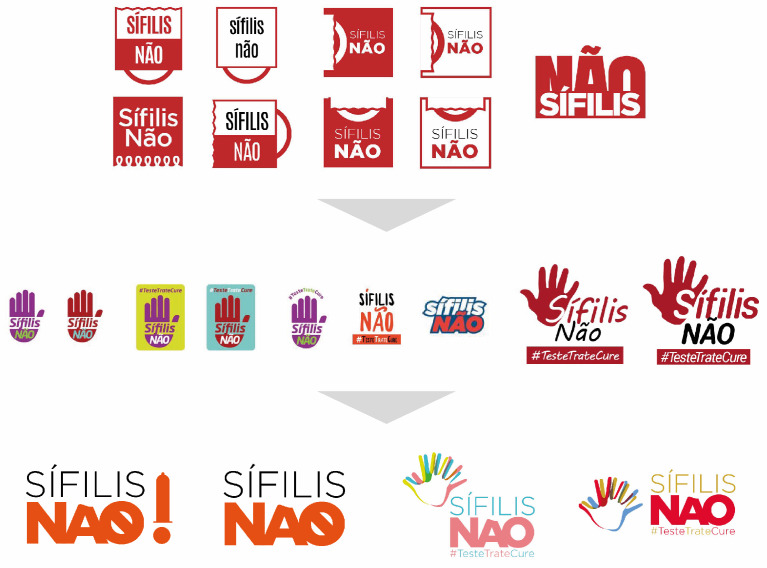
Process for creating the brand “Syphilis No”. **Source**: Projeto “Sífilis Não” [20].

**Figure 3 ijerph-19-15884-f003:**
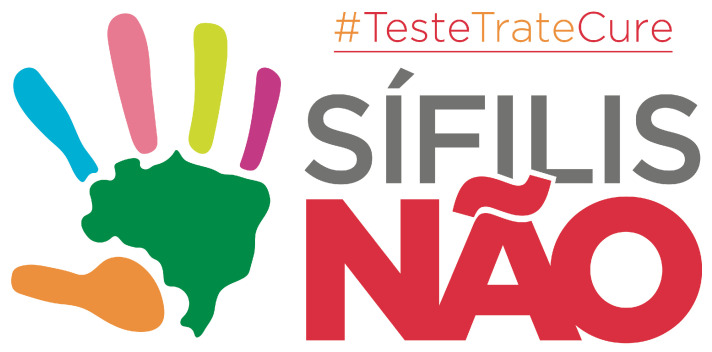
Brand of the project “No Syphilis” (Portuguese term “Syphilis No”). **Source**: Projeto “Sífilis Não” [20].

**Figure 4 ijerph-19-15884-f004:**
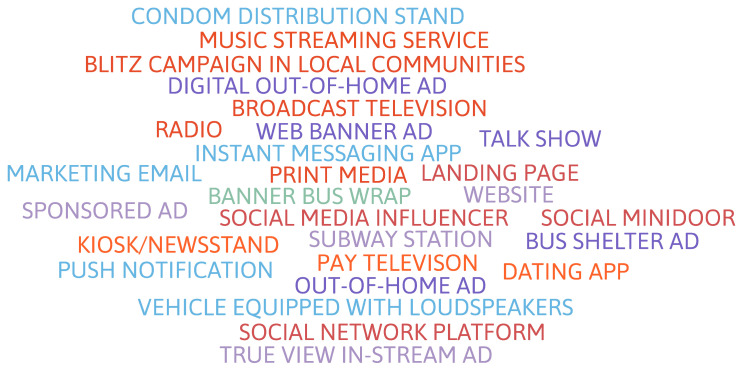
Word cloud of media and communication vehicles—Campaign 2.

**Figure 5 ijerph-19-15884-f005:**
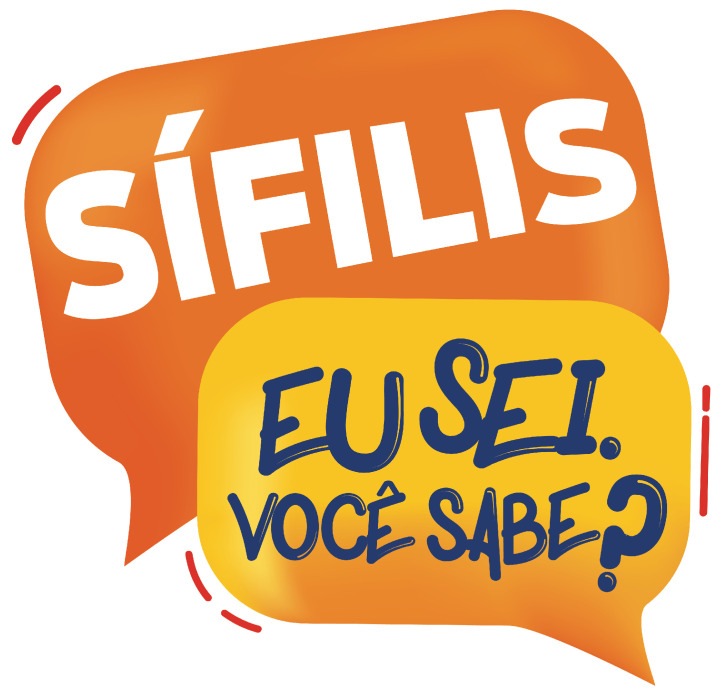
Visual identity, Campaign 3. **Source**: Projeto “Sífilis Não” [20].

**Figure 6 ijerph-19-15884-f006:**
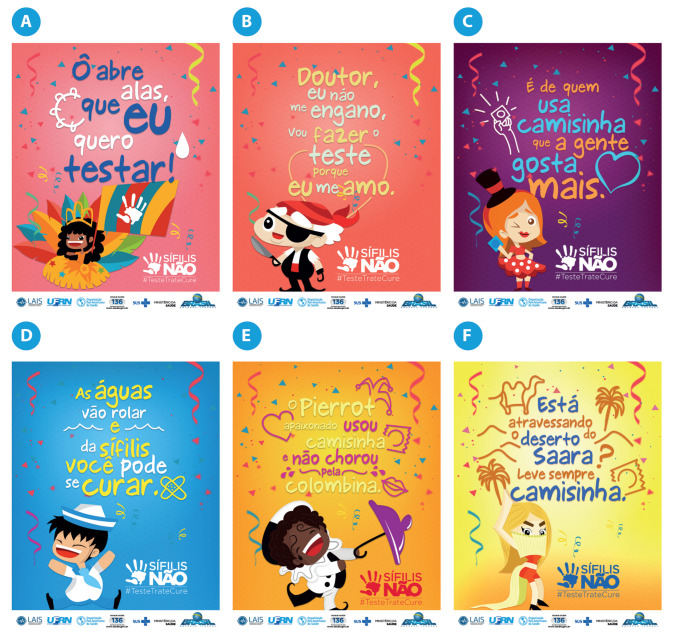
The elements developed for Campaign 1. **Source**: Projeto “Sífilis Não” [20].

**Figure 7 ijerph-19-15884-f007:**
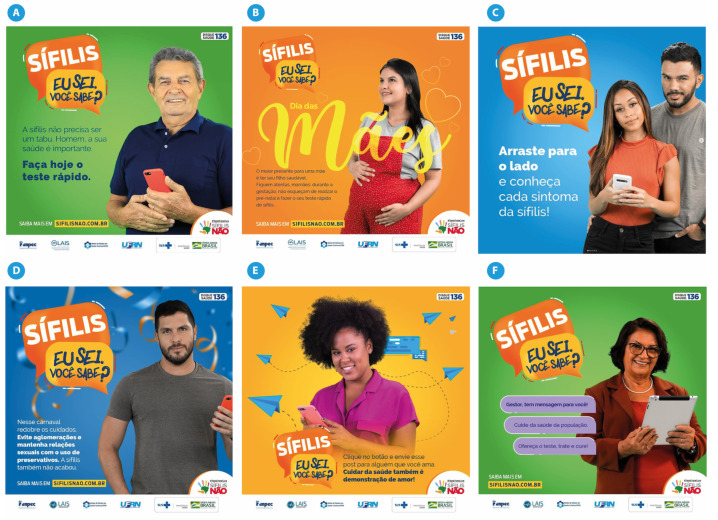
Campaign 3 material. **Source**: Projeto “Sífilis Não” [20].

**Figure 8 ijerph-19-15884-f008:**
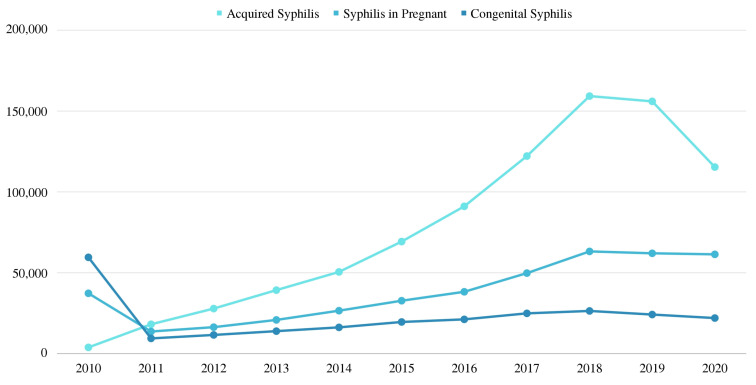
Cases of acquired syphilis, syphilis in pregnant women, and congenital syphilis (2015–2020).

**Figure 9 ijerph-19-15884-f009:**
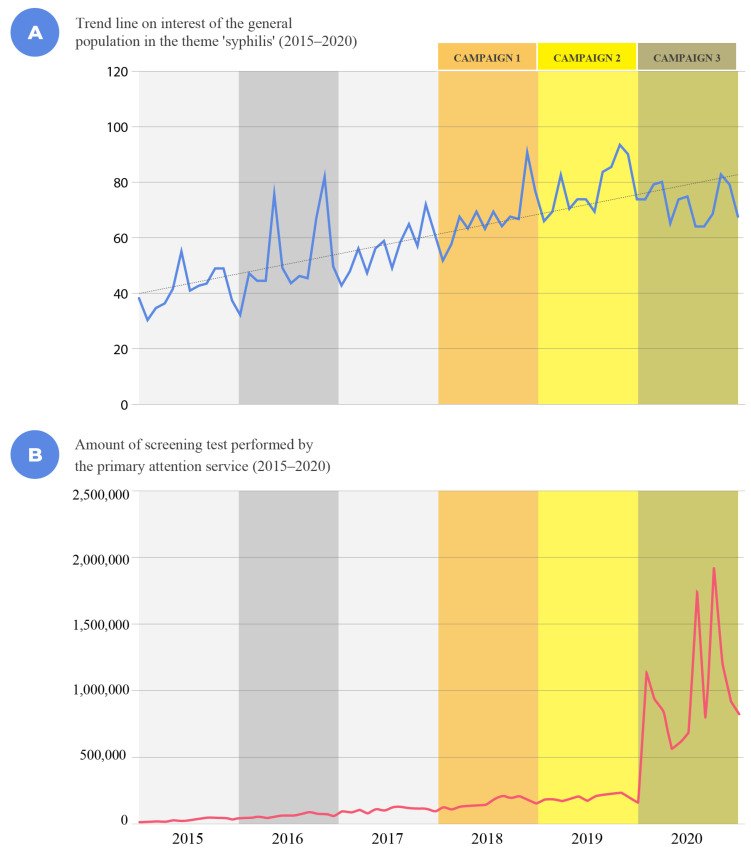
Trends in the interest of the general population in the term ‘syphilis’ (2015–2020).

**Table 1 ijerph-19-15884-t001:** Advertising campaigns carried out within the scope of the “Syphilis No” project.

Year	Theme	Focus	Target Audience
2018	“Syphilis No” #TestTreatCure	Carnival	Carnival, Key, and Priority Populations
2018–2019	“Remember to take care of yourself”	Prevention, testing, and treatment	General population and key and priority populations
2020–2021	“I know. Do you know?”	Prevention, testing, and treatment	General population and key and priority populations

**Source:** Adapted from [19].

**Table 2 ijerph-19-15884-t002:** Target audiences of the communication campaigns of the “Syphilis No” project.

Target Audience	Detailing
General population	-
Men	-
Managers	-
Health Professionals	-
Women/pregnant women	-
Sexual partners	-
Priority populations	Adolescents and young adults, Black people and people of color, Indigenous people, people experiencing homelessness.
Key populations	Persons deprived of liberty, people who use alcohol or drugs, gay people and MSM, transgender people, sex workers.

**Table 3 ijerph-19-15884-t003:** Visual identities produced based on the “Syphilis No” project.

City/State	Media Type	Description
Roraima/TO	Poster	Produced by the Basic Health Unit (UBS) team to raise awareness about protection against syphilis in the territory
São Paulo/SP	Manual materials	Material developed by health professionals to be delivered to UBS users during the strategies in the month of green October 2021
Pirituba/SP	Prize engraving	Award received for the green October campaign to combat congenital syphilis carried out in the region
Beryl/MG	Banner	To distribute the week’s strategies to combat syphilis to the health units of the municipality, which took place from 25 October to 29 October 2021. https://berilo.mg.gov.br/sifilis-nao-/, accessed on: 18 October 2022
Natal/RN	Shirts	Strategy promoted by Cidade Praia’s basic health team to raise awareness of syphilis

**Table 4 ijerph-19-15884-t004:** Description of the TV Linking of Campaign 2 of the “Syphilis No” Project.

Broadcaster	Channel Type	Binding Times (Breaks)	Public
Rede Globo	Open	Jornal Nacional, Hora Um, Encontro e Novelas	General
SBT	Open	SBT Brasil, A Praça é Nossa, Novela, Programa do Ratinho, Sábado Animado	General
Record	Open	Fala Brasil, Novela, Hoje em dia e Jornal da Record	General
Band	Open	Melhor da Tarde e Jogo Aberto	General
AMC Brazil	Closed	-	Young people (15 to 24 years old), gay men, MSM and men (20 to 35 years old)
HBO	Closed	-	Young people (15 to 24 years old), gay men, MSM and men (20 to 35 years old)

**Table 5 ijerph-19-15884-t005:** Results of Campaign 2. **Source**: Pinto et al. [2].

Communication Strategy	Number/Quantity	Unit of Measurement
Television	16,374,777	spectators reached
Print (two magazines)	10,800,000	impressions
Posters	193,500	distributed
Folders	770,000	distributed
Stickers	770,000	distributed
Radio + Streaming	4,141,527	audience reached
Video clip “Sem Vacilação”	150,000	views
Pamphlet + condoms distributed	38,000 pamphlets, 8000 condoms	people reached
Sound car	9440	hours
Street Furniture (1610 faces)	630,219,491	estimated impact (per week)
Digital Influencers	322,240,364	impact (on 10 digital social media accounts)

## Data Availability

Not applicable.

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
