# Peer review of "Analysis of the Impact of Communication Campaigns under the Project “Syphilis No”: A National Tool for Inducing and Promoting Health"

_ijerph, 2022, doi:10.3390/ijerph192315884_

Round 1

Reviewer 1 Report

This is an important piece of research considering the public health issue that is involved. 

The introductory part includes some sources from the literature - that could be improved by adding a more comprehensive review in order to highlight the research gap (e.g., previous studies that have addressed the same topic and from a similar/different research design would be really helpful to know). 

The methodological section could also be improved by providing a clear explanation of the methods of data analysis applied to the communication campaigns - at this stage, it seems to be a superficial description and not a critical analysis based on scientificaly proven tools.   

Lastly, a final section stating the limitations of the study and some avenues for further research would be appropriate.

Author Response

This is an important piece of research considering the public health issue that is involved. 

The introductory part includes some sources from the literature - that could be improved by adding a more comprehensive review in order to highlight the research gap (e.g., previous studies that have addressed the same topic and from a similar/different research design would be really helpful to know). 

Reply: Dear reviewer, we appreciate all the suggestions that enrich our work. To improve our work, we searched the literature and added some references in the introduction that support and strengthen the research theme. The added references are underlined in the text, they are: [3], [4], [5], [14], [18].

The methodological section could also be improved by providing a clear explanation of the methods of data analysis applied to the communication campaigns - at this stage, it seems to be a superficial description and not a critical analysis based on scientificaly proven tools.   

Reply: In the Methodology section, we added an excerpt detailing the data analysis methods applied to communication campaigns. Added snippet:

"When applying the process suggested by Cox in this research, the following steps were followed: (i) collection of data and epidemiological information on syphilis in Brazil; (ii) identification of official data from government agencies on syphilis in Brazil (Brazilian Ministry of Health and Brazilian Federal Court of Auditors); (iii) analysis of the data and information collected; (iv) identification of the problem, considering that the government carried out communication actions with the objective of combating syphilis, but the number of cases had grown; (v) mapping of different family/data sources on the research topic and; (vi) discussion about the research and its guiding question.

Thus, the "Sífilis Não" Project was classified as being considered the largest research project for inducing public policies to fight syphilis in Brazil. The project is composed of several thematic axes in relation to this health issue. It was decided to work on the communication axis. Among several actions, it was decided to analyze the public communication campaigns that were carried out in the Project, due to their behavioral familiarity and high level of detail. Public data on campaign formats makes it possible to replicate them for different locations or diseases"

Lastly, a final section stating the limitations of the study and some avenues for further research would be appropriate.

Reply: We also added a paragraph that addresses the main limitations and perspectives for further research. This paragraph is essential to indicate works that can continue this research. Follows the added section:

“As a result of innovative, transdisciplinary, intersectoral, and interfederative planning, it is possible to conclude that communication campaigns are effective health induction and promotion tools in the context of public health. However, campaigns cannot be stated as the only condition to explain the changes in trends for syphilis cases in Brazil. It is recommended to continue research in this area of study, expanding to other regions or diseases to investigate whether the methodology for disseminating information from the “No Síphilis” Project increases the interest of the general population and science on the subject and improves the scenario of health.”

Author Response

Thank you for the opportunity to read your article “Analysis of the Impact of Communication Campaigns under the Project “Syphilis No”: A National Tool for Inducing and Promoting Health”.

A study that exposes the development and results of a strategic intervention in response to a serious public health problem in a specific country that, due to its representativeness in the positive results achieved, can motivate similar interventions in other countries.

The research is well written, in clear and accessible language, with adequate articulation of the topics covered and with a solid and relevant scientific literature.

Title: It ́s concise, clearly indicates the content of the study and the approach used.

Abstract: Attractive, encourages reading, exposes the study's focus, objective, method, main results and what the reader can learn from reading.

Introduction: with clarity in the literature review, exposing in an objective, relevant and interconnected way the knowledge about the specific topic of the study; within a clear framework of the study context; the objective of the study is discussed, guiding questions are proposed and the objective of the study is explained.

Methods: The article is explicit in relation to the analysis process and in the description of data characteristics, sources used for extraction and analysis procedures.

I particularly liked the clear and succinct way in which the campaigns carried out, using the visualization of different graphic exhibitions.

Results: The results are very clear in describing the main characteristics of the campaign data and the relationships between them. However, in this section, results from other sources described in point 2.1 are not presented, realizing the authors' option to describe them in the discussion

Discussion: Very well in the purpose of the study that was present throughout the article.

Conclusion: The main conclusions in response to the objective outlined are presented, and were suggest future research based on the results found.

However, there are some aspects that I suggest changing:

Line 120: the description in the text seems to refer to figure 3;

Reply: This was done.

Line 124: the figure below this line, the last point 05 is not translated into English

Reply: This was done.

Lines 251-260: In the analysis of epidemiological indicators, only results are described, lacking an exposition of the analysis carried out, for example the contribution or implications in relation to the existing knowledge.

Reply: Dear reviewer, we appreciate all the suggestions that enrich our work. To enrich the analysis of the epidemiological indicators, we highlight in the text the influence of carrying out the campaigns on reducing cases of syphilis and increasing the search for the subject in search tools, such as Google. To this end, we have modified the following paragraphs:

As a complement to the analysis of the impact of the campaigns carried out, the epidemiological indicators of acquired syphilis, in pregnant women and congenital syphilis in Brazil showed an increasing number of cases since 2011, which began to change the trend from 2018 (Brasil, 2022). The number of cases of acquired syphilis, in 2018, reached an absolute total of 159,237. In 2020, that rate dropped to 115,371, representing a drop of 27.55%. Regarding congenital syphilis, the total value in 2018 corresponded to 26,464. In 2020, this figure also decreased to 22,065, representing a drop of 16.62%. The same pattern occurred with the number of syphilis cases in pregnant women, which in 2018 had 63,250 cases, and in 2020 this was 61,441, representing a drop of 2.86%. Figure 10 shows the number of cases of acquired syphilis, congenital syphilis and syphilis in pregnant women from 2010 to 2020. In this context, the contents of the campaigns carried out during this period may have reached a large part of the target population and, consequently, influenced the reduction of cases of syphilis in Brazil.

Another relevant aspect for analyzing the impact of the campaigns carried out is related to the investigation of content in web search engines.

Lines 266-267: Regarding the discussion about the data in Figure 11 (a) it is not clear what is described in this sentence

Reply: For better understanding, we added the following sentence: "The values shown in Figure 11A are not absolute numbers of searches." 

Lines 275- 280: What is intended to convey in these sentences is not clear to me.

Reply: For better understanding, we have modified the following paragraph:

“Another important point is the increase in the number of diagnostic tests performed. The study on the reach of public health campaigns based on multidimensional aspects by de Morais Pinto et al. [9], considers these indicators of the performance: serological tests for syphilis, drug distribution (Penicillin G benzathine) and case notification rates in Primary Health Care (PHC). All these data are from 2015 to 2019, based on data collected from SIA/SUS database. In addition, the data equivalent to the year 2020, year of the last communication campaing analyzed, was also taken from SIA/SUS database. These data collected were relating it to population data, calculating its rate per 100,000 inhabitants - to compare the years data in the same line base of analysis - and plotted in Figure 11B.”

Thank you.

Reviewer 3 Report

Dear authors, I congratulate you for focusing on a subject with an important social function and for understanding that communication campaigns, well created and planned, are useful in the public health promotion strategy. I suggest you improve the paper by better organizing and simplifying the information and data of the campaigns, using figures and tables properly, to understand the work and the results.

Author Response

Dear authors, I congratulate you for focusing on a subject with an important social function and for understanding that communication campaigns, well created and planned, are useful in the public health promotion strategy. I suggest you improve the paper by better organizing and simplifying the information and data of the campaigns, using figures and tables properly, to understand the work and the results.

Reply: Dear reviewer, we thank you for your comment. At the same time, we are happy to corroborate with scientific research that reinforces the importance of communication in health for a better quality of life for people. Improvements throughout the work were made for better understanding.

Reviewer 4 Report

de Lima Paiva et al. In their manuscript entitled ‘Analysis of the Impact of Communication Campaigns under the Project “Syphilis No”: A National Tool for Inducing and Promoting Health’ describe a series of 3 public awareness/health campaigns aimed at reducing syphilis cases in Brazil with seemingly quite a positive impact. These types of public health campaigns are hugely important with both new and re-emerging communicable diseases. The description of the campaigns is quite all encompassing having reached out to a very large, almost universal, part of the Brazilian population. My major criticism is the organisation of the manuscript in that the discussion has very little actual discussion or analysis and mostly contains linked results of the campaign impact. The conclusion on the other hand does contain the start of a good discussion. Also, lengthy sentences and grammatical errors often make the overall point of the authors difficult to understand.

 Minor comments :

-the link in reference 4 is broken and thus I cannot verify the statistics provided in the text. I believe these are absolute cases not per 100,000 inhabitants, 1,000 births etc.?

-who were the key populations identified in Judgment No. 2019/2017? This will be important in assessing the accessibility of the educational intervention. It does feature in the methods later on but would be good to highlight these groups in the introduction.

-what is meant by “what family does the data come from”?

-line 165 I think you mean ‘verses’ not versus?

-would suggest removing the word ‘gays’ as this is encompassed in the term MSM or use the term ‘gay men’

-I am unclear what this sentence means “This campaign stands out for the fact that personas (the target audience of the campaign) are not actors, individuals are in real life what they are representing in the images.”

-black populations are mentioned twice in lines 224-225 as are health workers

-“This factor collaborates to justify…” either the word collaborate needs to be removed or perhaps the authors mean corroborate?

 Major comments:

-the introduction contains content that should be in the results and discussion section and seems misplaced eg. Google search results and discussion on assumptions

-I am unclear what the authors mean by the following statement: “Finally, an individual analysis of the data obtained was carried out and, then, a descriptive analysis of the information in an integrated methodology”?

-Why was campaign 1 NOT fully applied or adapted in these areas if it was a national campaign? “Table 2 describes examples of how the Syphilis Project visual identity was not fully applied or adapted in the states of São Paulo, Tocantins, Minas Gerais and Rio Grande do Norte.”

-there is considerable repetition in the description of the campaigns in the methods, then again at the start of each campaign paragraph in the results section

-the syphilis case data pre and post campaign should be presented in the results section. Then the causative analysis on the drop in numbers should be discussed in the discussion section and the authors can link this to the 3 campaigns that were implemented. The same goes for Google searches and testing data.

-The conclusion should be the main points in the discussion. In the discussion the authors could also include things such as how this experience and the campaign developed could be expanded to include other communicable diseases of great importance, a reflection on what could have been changed to improve an already successful campaign, and an economic impact statement (ie. cost of delivering the campaigns vs healthcare cost savings) would be very interesting and an important aspect of this analysis

Author Response

de Lima Paiva et al. In their manuscript entitled ‘Analysis of the Impact of Communication Campaigns under the Project “Syphilis No”: A National Tool for Inducing and Promoting Health’ describe a series of 3 public awareness/health campaigns aimed at reducing syphilis cases in Brazil with seemingly quite a positive impact. These types of public health campaigns are hugely important with both new and re-emerging communicable diseases. The description of the campaigns is quite all encompassing having reached out to a very large, almost universal, part of the Brazilian population. My major criticism is the organisation of the manuscript in that the discussion has very little actual discussion or analysis and mostly contains linked results of the campaign impact. The conclusion on the other hand does contain the start of a good discussion. Also, lengthy sentences and grammatical errors often make the overall point of the authors difficult to understand.

 Minor comments :

-the link in reference 4 is broken and thus I cannot verify the statistics provided in the text. I believe these are absolute cases not per 100,000 inhabitants, 1,000 births etc.?

Reply: 

Dear reviewer, we appreciate your valuable comments that have helped to improve our article.

This was done. The link has been updated. Thank you for your attention. We update the data according to the link:

“In Brazil, according to official data from the Epidemiological Bulletin of the Ministry of Health of Brazil (MH), in 2020, the incidence rate (per 100,000 habitants) of acquired syphilis was 54,5; 21,6 of syphilis in pregnant women (per 1,000 live births); and 7,7 of congenital syphilis in children under one year old (per 1,000 live births)”

- who were the key populations identified in Judgment No. 2019/2017? This will be important in assessing the accessibility of the educational intervention. It does feature in the methods later on but would be good to highlight these groups in the introduction.

Reply: We add the following details in paragraph 4 of the Introduction:

“This audit resulted in Judgment No. 2019/2017 (Brasil, 2017), which recommended new and diversified health communication actions, targeted to specific audiences, for example: teenagers, sex workers, the population without internet access. The other TCU recommendations for effective communication and control of syphilis in Brazil include other audiences such as health professionals, managers, pregnant women, sexual partners, etc.”

-what is meant by “what family does the data come from”?

Reply: Dear reviewer, we apologize for the inconvenience. “what family does the data come from” is one of the questions/steps of the process suggested by Cox [19] to apply the experimental method of data analysis. For better understanding, the following excerpt has been added to the topic Materials and Methods:

"When applying the process suggested by Cox [19] in this research, the following steps were followed: (i) collection of data and epidemiological information on syphilis in Brazil; (ii) identification of official data from government agencies on syphilis in Brazil (Brazilian Ministry of Health and Brazilian Federal Court of Auditors); (iii) analysis of the data and information collected; (iv) identification of the problem, considering that the government carried out communication actions with the objective of combating syphilis, but the number of cases had grown; (v) mapping of different family/data sources on the research topic and; (vi) discussion about the research and its guiding question."

-line 165 I think you mean ‘verses’ not versus?

Reply: This was done.

-would suggest removing the word ‘gays’ as this is encompassed in the term MSM or use the term ‘gay men’

Reply: In Brazilian sexual culture, these issues generate ambivalence, considering that a man can have sex with another man and not be or not identify himself as homosexual. Therefore, the communication thought questions about gender and sexualities, specifically, around the distinction between sexual identity and sexual role. Thus, the Brazilian Ministry of Health recommended, as different categories, that the communication campaigns be carried out to seek to reach gays and men who have sex with men, considering that they may be audiences with varying cultures of consumers and that there should be efficient communication strategies to achieve both.

The category of “men who have sex with men (MSM)” is considered one of the priority populations for preventing STD/HIV/AIDS in Brazil.

-I am unclear what this sentence means “This campaign stands out for the fact that personas (the target audience of the campaign) are not actors, individuals are in real life what they are representing in the images.”

Reply: Dear reviewer, we apologize for the inconvenience. For better understanding, we have modified the following paragraph: 

"For example, in Figure 9C, which aimed to communicate to young couples, it is a real life couple; in Figure 3D it is a gay man to achieve communication with other gay men; in Figure 9F, with text aimed at managers in the health area, there is the image of a health manager in real life with notorious recognition for her actions to combat Sexually Transmitted Infections."

-black populations are mentioned twice in lines 224-225 as are health workers

Reply: This was done.

-“This factor collaborates to justify…” either the word collaborate needs to be removed or perhaps the authors mean corroborate?

Reply: We apologize for the misunderstanding. We've updated the word to "corroborates".

 Major comments:

-the introduction contains content that should be in the results and discussion section and seems misplaced eg. Google search results and discussion on assumptions

Reply: Thanks for the observation. We reformulated the last two paragraphs of the introduction and relocated the part that addresses the results to an acceptable section.

Reworded excerpt:

“In the field of public health communication, especially for syphilis awareness, social campaigns are acknowledged as a promising strategy that can lead to improved health by changing the behavior of individuals [16,17]. In addition, communication can be applied as an intrinsic strategy to health promotion, being considered an important public health intervention tool capable of inducing the prevention and control of infectious diseases [18]. Thus, this study starts from the hypothesis that the three communication campaigns developed by the “Sífilis Não” project had a positive and considerable impact on the syphilis scenario in the national territory in the period from 2018 to 2020. In order to guide the research, the following questions were defined: (i) Did the communication campaigns of the “Sífilis Não” project stimulate the interest of the general population on the subject? (ii) Did the campaigns induce an increase in testing?

Given the context addressed, this study aims to describe the impact of public health communication campaigns of the “Sífilis Não” project to approach syphilis in Brazil and analyze how these campaigns contributed to inducing behavior change that reflected in the improvement of syphilis indicators in Brazil, considering the data from 2015 to 2020.”

-I am unclear what the authors mean by the following statement: “Finally, an individual analysis of the data obtained was carried out and, then, a descriptive analysis of the information in an integrated methodology”?

Reply: Your comment helped us to improve the survey methodology. We improved the detailing to make it clearer.

-Why was campaign 1 NOT fully applied or adapted in these areas if it was a national campaign? “Table 2 describes examples of how the Syphilis Project visual identity was not fully applied or adapted in the states of São Paulo, Tocantins, Minas Gerais and Rio Grande do Norte.”

Reply: The three campaigns were nationwide. We have improved the description of this part for better understanding.

"The communication campaigns of the “Sífilis Não” project have different characteristics. Campaign 1 is fundamental to establish an identity against syphilis in Brazil. The brand created by the researchers of the “Sífilis Não” Project was used as an icon in actions to combat the disease in several regions of the country and became a symbol of combating syphilis in Brazil.For example, the red ribbon symbol is used to talk about HIV/AIDS.

Entities with no direct link to the Project use the brand in Figure 2 in their own actions. Table 2 describes examples of how the Syphilis Project visual identity was not fully applied or adapted in the states of São Paulo, Tocantins, Minas Gerais and Rio Grande do Norte in state actions to communicate about syphilis."

-there is considerable repetition in the description of the campaigns in the methods, then again at the start of each campaign paragraph in the results section

Reply: We appreciate your note. Improvements have been made throughout the text. Some repetitions can still be identified and have been kept for better understanding by readers.

-the syphilis case data pre and post campaign should be presented in the results section. Then the causative analysis on the drop in numbers should be discussed in the discussion section and the authors can link this to the 3 campaigns that were implemented. The same goes for Google searches and testing data.

Reply: We understand your point of view, but we would like to clarify that the data related to syphilis cases in Brazil before and after the campaign are presented in the article as complementary data in order to enrich the discussion on the results obtained in the campaigns carried out in several years. Thus, the main objective of this article is to present the advertising campaigns and discuss their impacts related to the fight against syphilis in Brazil. Therefore, we appreciate your suggestion, but we consider it more appropriate to keep this data in the discussion section.

To enrich this analysis, we emphasize in the text the possibility of the influence of campaigns in view of the reduction of syphilis cases and the increase in demand on the subject in research tools, such as Google. With that, we modify the following paragraphs:

“As a complement to the analysis of the impact of the campaigns carried out, the epidemiological indicators of acquired syphilis, in pregnant women and congenital syphilis in Brazil showed an increasing number of cases since 2011, which began to change the trend from 2018 (Brazil , 2022). The number of cases of acquired syphilis, in 2018, reached an absolute total of 159,237. In 2020, that rate dropped to 115,371, representing a drop of 27.55%. Regarding congenital syphilis, the total value in 2018 corresponded to 26,464. In 2020, this figure also decreased to 22,065, representing a drop of 16.62%. The same pattern occurred with the number of syphilis cases in pregnant women, which in 2018 had 63,250 cases, and in 2020 this was 61,441, representing a drop of 2.86%. Figure 10 shows the number of cases of acquired syphilis, congenital syphilis and syphilis in pregnant women from 2010 to 2020. In this context, the contents of the campaigns carried out during this period may have reached a large part of the target population and, consequently , influenced the reduction of cases of syphilis in Brazil.

Another relevant aspect for analyzing the impact of the campaigns carried out is related to the investigation of content in web search engines.”

However, it is still valid to emphasize that it is not feasible to say that campaigns are the only condition to explain changes in trends in syphilis cases in Brazil. At the conclusion of the article, we recommend the continuity of research in this area of ​​study to verify the causality between the analyzed indicators.

-The conclusion should be the main points in the discussion. In the discussion the authors could also include things such as how this experience and the campaign developed could be expanded to include other communicable diseases of great importance, a reflection on what could have been changed to improve an already successful campaign, and an economic impact statement (ie. cost of delivering the campaigns vs healthcare cost savings) would be very interesting and an important aspect of this analysis

Reply: In order to expand the discussion, we modified some parts and added a paragraph that addresses the main limitations of the study and which recommendations for further studies corroborate our results. In fact, the economic impact is a very relevant topic, however, due to its scope and complexity in the Brazilian Unified Health System, we emphasize that further research should be carried out to characterize and analyze investments in this area.

The following excerpt was added to the Conclusion:

“From the research carried out, relevant results were found. It is possible that there is a relationship between the innovative, transdisciplinary, intersectoral and interfederative way that the campaigns were carried out with the improvement of the epidemiological scenario of syphilis in Brazil. However, causality cannot be stated. It is recommended to continue research in this area of ​​study, expanding to other regions or diseases to investigate whether the methodology for disseminating information from the “No Síphilis” Project increases the interest of the general population and science on the subject and improves the scenario of health.”

Reviewer 5 Report

In this paper, the influence of the propaganda activities carried out by the "syphilis-free" project is studied. This topic has a certain degree of significance in popularizing the knowledge of "syphilis" in state institutions, guiding people to know more about related knowledge and promoting social health construction, and has certain academic research value. However, some defects or problems in the revised paper still need to be improved, as follows:

1.       Data related to publicity activities need more explanation. Although this paper explains several kinds of publicity activities in detail, it does not explain more about some necessary statistical data involved in the publicity activities, which will lead to the lack of sufficient data support in the analysis of the results.

2.       The fourth part of this paper needs more powerful elaboration and analysis, in order to provide support for the research conclusion. Although the fourth part of this article explains the changing trend of the search volume of "syphilis" in Google's Google entry, it doesn't clearly explain the relationship between the propaganda activities and the change of the search volume of "syphilis" in Google search. In addition, there are many reference indicators that can be selected, but this article does not explain the reasons for choosing relevant data searched by Google as reference indicators.

3.       There are many factors influencing the number of registered cases of syphilis in Brazil. In the fourth part of this paper, we simply explain that the number of cases is affected by the development of related projects, but there is no explanation or comparative analysis of other factors that affect this change.

4.       The relationship between OBIS, ATAS, personal characteristics and experimental results in the questionnaire components used in the experiment of this paper needs to be explained.

5.       The elaboration and explanation of the conclusion in this paper are not sufficient.

Author Response

In this paper, the influence of the propaganda activities carried out by the "syphilis-free" project is studied. This topic has a certain degree of significance in popularizing the knowledge of "syphilis" in state institutions, guiding people to know more about related knowledge and promoting social health construction, and has certain academic research value. However, some defects or problems in the revised paper still need to be improved, as follows:

1. Data related to publicity activities need more explanation. Although this paper explains several kinds of publicity activities in detail, it does not explain more about some necessary statistical data involved in the publicity activities, which will lead to the lack of sufficient data support in the analysis of the results.

Reply: In fact, it is a very important suggestion for the enrichment of the project, since the "No Syphilis" Project, developed by the Federal University of Rio Grande do Norte (UFRN), has as its main characteristic the development of research aimed at fighting syphilis in Brazil. The statistical data referring to the advertising campaign itself are the scope of another research that focuses on greater detail and analysis for this purpose. This study aims to describe and analyze the impact of the public health communication campaigns of the “No Syphilis” Project on the approach to syphilis in Brazil and to analyze how these campaigns contributed to inducing changes in behavior that reflected in the improvement of syphilis indicators in Brazil, considering data from 2015 to 2020; however, it is not possible to affirm causality and it is recommended to continue studies in the area to verify.

2. The fourth part of this paper needs more powerful elaboration and analysis, in order to provide support for the research conclusion. Although the fourth part of this article explains the changing trend of the search volume of "syphilis" in Google's Google entry, it doesn't clearly explain the relationship between the propaganda activities and the change of the search volume of "syphilis" in Google search. In addition, there are many reference indicators that can be selected, but this article does not explain the reasons for choosing relevant data searched by Google as reference indicators.

3. There are many factors influencing the number of registered cases of syphilis in Brazil. In the fourth part of this paper, we simply explain that the number of cases is affected by the development of related projects, but there is no explanation or comparative analysis of other factors that affect this change.

Reply: In order to improve readers understanding, it was added in 'Conclusion' that this research still cannot affirm causality between the realization of the campaigns of the "No Syphilis" Project and the improvement of the syphilis scenario in Brazil. Future research is recommended.

Still, it was added in the second part of this article when the choice to consider data from Google searches to identify the interest of the general population in seeking more information about syphilis, considering the periods in which the campaigns were carried out and also aimed at the dissemination information and stimulus for the constant search for knowledge.

Here's the adjusted snippet:

"(iii) and the Google Trends tool to gauge the population's interest in 'syphilis', that is, in the search for more knowledge and information on the theme".

In this way, we answer questions 2 and 3.

4. The relationship between OBIS, ATAS, personal characteristics and experimental results in the questionnaire components used in the experiment of this paper needs to be explained.

Reply: We apologize, but we do not recognize what was described as part of our article.

5. The elaboration and explanation of the conclusion in this paper are not sufficient.

Reply: In perspective to expand the discussion, we modified some parts and added a paragraph that addresses the main limitations of the study and what recommendations for further studies corroborate our results.

Round 2

Reviewer 1 Report

I am happy with the amendments and additions, many thanks

Author Response

Dear reviewer, thank you very much for your comments. We are sure that they were very significant for the enrichment of our work.

Reviewer 3 Report

The improvement in the structuring of the information seems significant, especially with regard to the graphic pieces of the advertising campaigns, which favours the monitoring and presentation of results. However, I consider that, for a scientific journal of the quality of IJERPH, the infographics in figure 4 and figure 8 should be improved. I suggest that they create their own table with this information, and that it should not be a "cut and paste" of an existing infographic.  With this change, everything is improved. Thank you!

Author Response

The improvement in the structuring of the information seems significant, especially with regard to the graphic pieces of the advertising campaigns, which favours the monitoring and presentation of results. However, I consider that, for a scientific journal of the quality of IJERPH, the infographics in figure 4 and figure 8 should be improved. I suggest that they create their own table with this information, and that it should not be a "cut and paste" of an existing infographic.  With this change, everything is improved. Thank you!

Reply: Dear reviewer, we reiterate our thanks for your review to improve the article. Information from figures 4 and 8 has been placed in tables 2 and 5. The numbers of other figures and tables and their mentions in the text have been updated.

Reviewer 4 Report

Thank you very much asking me to re-view this updated manuscript. Considerable improvements have been made in this draft and as a result the flow is much better and the narrative developed presented in a much more logical sequence.

I would suggest that that the first paragraph in the conclusion be moved into the discussion. I am still confused about the sentences in line 174-179 that state the campaign was not fully applied in all states yet the authors describe this as a national campaign in their response to previous comments.  Perhaps this is a linguistic oversight?

Before publication, I would suggest considerable linguistic review in order to ensure the main messages are communicated in a manner that is comprehensible to the reader and are not misinterpreted.

Author Response

Thank you very much asking me to re-view this updated manuscript. Considerable improvements have been made in this draft and as a result the flow is much better and the narrative developed presented in a much more logical sequence.

I would suggest that that the first paragraph in the conclusion be moved into the discussion. I am still confused about the sentences in line 174-179 that state the campaign was not fully applied in all states yet the authors describe this as a national campaign in their response to previous comments.  Perhaps this is a linguistic oversight?

Reply: Dear reviewer, we reiterate our thanks for your collaboration in improving our article.

The new suggestions were accepted. The first paragraph of the conclusion was moved to discussion and the text of lines 169-176 was improved to: 

"The brand created, in campaign 1, by the researchers of the Project “Sífilis Não” was used as an icon in actions to combat the disease throughout the country and became a symbol of the fight against syphilis in Brazil. It is also used by people who are not part of the project. For example, how the red ribbon symbol is used by anyone to talk about HIV/AIDS.

Entities with no direct link to the Project used the brand in Figure 2 in their own actions. Table 2 describes some examples of how the visual identity created for the national campaign of the Project "Syphilis No" was fully applied or adapted in the states of São Paulo, Tocantins, Minas Gerais and Rio Grande do Norte in state communication actions on syphilis. This means that the brand created for the national campaign was replicated by other people and became an element that represents the fight against syphilis."

Before publication, I would suggest considerable linguistic review in order to ensure the main messages are communicated in a manner that is comprehensible to the reader and are not misinterpreted.

Reply: In addition, as requested, we performed a linguistic review of the entire text.